# Providing Alternative Hosts and Nectar to Aphid Parasitoids in a Plum Orchard to Determine Resource Complementarity and Distance Range Effect on Biological Control

Louise Lérault [1,2], Elsa Clavel [2], Cinthya M. Villegas [2], Nuri Cabrera [2], Bruno Jaloux [1], Manuel Plantegenest [3] and Blas Lavandero [2,*]

1    IGEPP, INRAE, Institut Agro, Univ. Rennes, 49045 Angers, France; louise.lerault@agrocampus-ouest.com (L.L.); bruno.jaloux@agrocampus-ouest.fr (B.J.)
2    Laboratorio de Control Biológico, Instituto de Ciencias Biológicas, Universidad de Talca, Talca 3465548, Chile; elsa.clavel@agrocampus-ouest.fr (E.C.); cvillegas@utalca.cl (C.M.V.); ncabrera@utalca.cl (N.C.)
3    IGEPP, INRAE, Institut Agro, Univ. Rennes, 35000 Rennes, France; manuel.plantegenest@agrocampus-ouest.fr
*    Correspondence: blavandero@utalca.cl

**Abstract:** There are many different practices that contribute to conservation biological control, but little is known about their complementarity. We tested the effects of providing food and alternative hosts to parasitoids by intercropping a plum orchard with companion plants. Oats and vetch were intercropped into the orchard either as single-species (oats or vetch) or two-species (oats and vetch combined) intercrops within an inter-row. The trophic resources provided by these intercrops were assessed, along with the incidences of aphids and their parasitoids in plum trees. We found up to ten alternative host species provided by oats and vetch, and extrafloral nectar was available from the vetch and mixed strips. An effect of intercrop type and distance to plum trees was observed on aphid incidence during one sampling period. Parasitism rates in exclusion cages were affected by intercrop type, reaching almost 60% close to the mixed intercrop. However, no general tendency was observed upon whether oats, vetch or their mixture was associated with a lower incidence of aphids. We found no evidence that providing effective sources of food and alternative hosts for parasitoids increased aphid mortality in this study. The context-dependent efficiency of intercropping is discussed.

**Keywords:** orchard intercropping; alternative hosts; extrafloral nectar

## 1. Introduction

The trophic resources of most living organisms are heterogeneously distributed in time and space [1]. Regardless of the spatial or temporal scale considered, the resources used by an organism vary in diversity and abundance, and their distribution influences the organism's behavior [2]. For aphid parasitoids such as Aphidiinae [Hymenoptera: Braconidae], whose adults lay eggs in aphids and can feed on various carbohydrate sources, aphid hosts and carbohydrates are two essential and complementary trophic resources [3,4]. Indeed, the reproductive success of the parasitoids depends on the availability of both resources. Hence, the spatial and temporal distributions of these resources strongly influence the local density of parasitoids and their control of the host population.

Parasitoids use aphid hosts primarily for reproduction and overwintering [4,5]. However, the availability of aphid hosts in cropping systems varies throughout the year in temperate regions. Accordingly, parasitoids are active as biocontrol agents especially during the warm season, when pest aphid populations start to grow within the crop. The diversification of aphid species during evolution has generated diverse plant–host–parasitoid tritrophic interactions, defining host preferences within parasitoid communities [4,6]. Yet, this specialization process is generally not restricted to one host species, and parasitoids may exploit various aphid hosts on various plants, less preferred host species

being considered as alternative hosts [7,8]. Therefore, increasing plant diversity within an agroecosystem could lead, through the provision of alternative hosts for parasitoids, to an increase in their early population and their maintenance when pest aphids are scarce. Due to their development time, parasitoids do not control aphid populations immediately, unlike predators, which means that early arrival of parasitoids is necessary to ensure a high level of efficiency in pest control [9,10]. Plant and host diversities are then of great importance as they affect parasitoid survival, the timing of their colonization of agricultural fields in spring, and the size of their populations [11–13].

At the same time, plant diversification should increase the diversity of carbohydrate sources for natural enemies, including honeydew produced by aphids and other insects such as psyllids or mealybugs, and plant-provided floral and extrafloral nectars [14]. Honeydew is available and accessible for parasitoids wherever aphids are located, but its quantity depends on aphid density. However, honeydew is of low nutritional value compared to plant nectar for parasitoids, who tend to choose locations where they can find nectar close to their hosts [15–17]. Nectar improves parasitoid longevity, female fecundity and parasitism activity [18–20]. However, nectar availability is limited in time and space, because not all plants produce accessible nectar, especially extrafloral nectar, and floral nectar is restricted to the flowering period [21–23]. Hence, enhancing nectar provisioning is encouraged when designing biological control programs to attract and retain parasitoids [24–27].

Understanding how parasitoids respond to temporal and spatial disparities in the availability of their trophic resources is a key point for improving the ecosystem service of pest regulation. As hosts and nectar have complementary positive effects on parasitoids, an additive effect on parasitoid populations and their effectiveness as aphid regulators can be expected when both resources are provided. However, the two resources rarely occur in the same location. Because of their small size, parasitoids have low energy reserves and must frequently find sugar sources, ideally near their hosts [28,29]. Although they are known for their ability to travel large distances (up to 20 m from the release point for *Lysiphlebus cardui* [30]), the positive impact of floral strips placed at the edge of crops on pest control is often limited to a few meters [17]. Therefore, the distance between hosts and nectar sources has an important impact on their activity [31–34].

There are different ways to provide resources in orchards, and several types of resources can be provided, either complementary (nectar, refuges) or supplementary (alternative hosts) with the targeted agronomic pest host [35–37]. Banker plants are used to provide alternative hosts to parasitoids in order to increase or maintain their populations when the target host pest is scarce, whereas nectariferous plants are useful to feed them when the crop of interest does not provide nectar. This is the case for many fruit trees that have a short flowering period, do not produce extrafloral nectar or provide it when parasitoids are not present or when it is almost exclusively used and defended by ants [23,38–40]. In orchards, well-chosen companion plants sown as intercrops in the inter-rows of fruit trees should have a positive effect on parasitoid populations and pest control because not only will high-quality resources be directly provided in the orchard, but these will be provided in close proximity to the pest (host). However, intercrops might also keep parasitoids within the inter-rows, where they have all the resources they need, and keep them from moving to another vegetation level [41], and adversely affect pest control.

Comparison of the effect of providing the two types of trophic resources in an orchard and the evaluation of their possible complementarity have never been carried out. Aphids can be serious pests for many fruit trees around the world, causing direct and indirect damage. In plum tree orchards, highly polyphagous aphid species such as *Myzus persicae* (Sulzer) [Hemiptera: Aphididae] or specialized species such as *Brachycaudus helichrysi* (Kaltenbach) [Hemiptera: Aphididae] can cause significant economic losses by stunting shoots, reducing plant growth, and transmitting viruses [42–45]. Planting carefully selected intercrops between rows of trees in a plum orchard could be a way to provide an alternative

aphid species for parasitoids when pest densities are low and nectar sources for parasitoids to effectively reduce the incidence of aphid pests on plum trees [46,47].

Cereals such as oats (*Avena sativa* L.) have already proven to be good banker plants to provide alternative hosts to parasitoids in orchards [48–50]. Indeed, none of the aphid species known to feed on oats includes plum as a host plant, but they do have parasitoid species in common with aphid species that attack plum [51,52]. These parasitoid species were presumably able to transfer from oats as an intercrop to orchard trees. Aphids on oats usually arrive early in the season, before plum trees start budding, which can be favourable for building up early parasitoid populations in the orchard [53,54].

Leguminous plants such as some vetches (*Vicia sativa* L., *V. benghalensis* L.) produce extrafloral nectar known to be consumed by aphidiine parasitoids [25,55]. As an atmospheric nitrogen-fixing legume, vetch does not compete for nitrogen with plum trees [56]. Furthermore, vetch growth in inter-rows does not interfere with normal orchard management (such as the transit of tractors or other machinery), making it a good nectariferous plant to provide extrafloral nectar in an orchard. Additionally, aphid species feeding on leguminous plants such as vetch arrive later in the season compared with oats, do not attack plum trees, and share parasitoid species with plum aphids [57].

In this study, we evaluated whether intercropping in a plum tree orchard could provide both types of trophic resource needed by parasitoids: early alternative host aphids through intercropping with cereals (oats), and extrafloral nectar with late alternative host aphids through intercropping with legumes (vetch). Additionally, we assessed whether intercropping increases parasitoid activity in plum trees. Along with deciphering the effect of each resource type on aphid regulation and parasitism activity, we hypothesized that, when composed of mixed species, intercrops would have a complementary effect and produce a higher impact on aphid regulation by parasitoids, either through additive or synergistic effects. We also estimated the spatial extent of the effect of these intercropping arrangements by measuring aphid incidence and parasitism activity in plum trees located at different distances from an intercropped inter-row. We hypothesized that the effect on parasitism and aphid incidence in plum trees will decline with the distance from the intercropped strips.

## 2. Materials and Methods

### 2.1. Study Site

The experiment took place in 2019, during a single growing season in an organic plum tree orchard located inside the farm "Alto Las Delicias" near the city of Rancagua, in the O'Higgins region of central Chile (34°08′07.6″ S 70°38′59.5″ W). A total area of about 3 hectares within the orchard was used for the experiment with plum trees of the cultivar 'd'Agen' of similar age and size, representing approximately one-third of the total plum orchard area (Figure 1). The management of the plum trees was the same throughout the experimental area.

### 2.2. Experimental Design

The central inter-row of the orchard area considered for the experiment was sown with three different types of intercrop patches: oats (*Avena sativa* L.), vetch (a mixture of 20% *V. sativa* and 80% *V. benghalensis*), and a mixture of both previous types (oats+vetch, same plant density), which will be referred to as the mixed intercrop (Figure 1). The mixed intercrop was sown using 50% less seeds of oats and vetch compared with the single crop patches. For all patch types, oats were mechanically sown in rows (around 20 cm of distance between rows, every other row not sown in the mixed patches). Vetches were also sown in rows but manually (same inter-row distance as oats). In the mixed intercrop, vetches were sown to fill in the missing rows. Each patch type covered a strip 20 m long, 5 m wide between two rows of plum trees, corresponding to a strip of 7 plum trees. Thirty meters were left unsown on both sides of every intercrop patch. The three patch types constituted a block, replicated three times along the inter-row (Figure 1). All the other inter-rows were

left unsown. Throughout the season, all flowering weeds growing in the area were mown to avoid the presence of nectar sources or aphid hosts other than the intercrop patches.

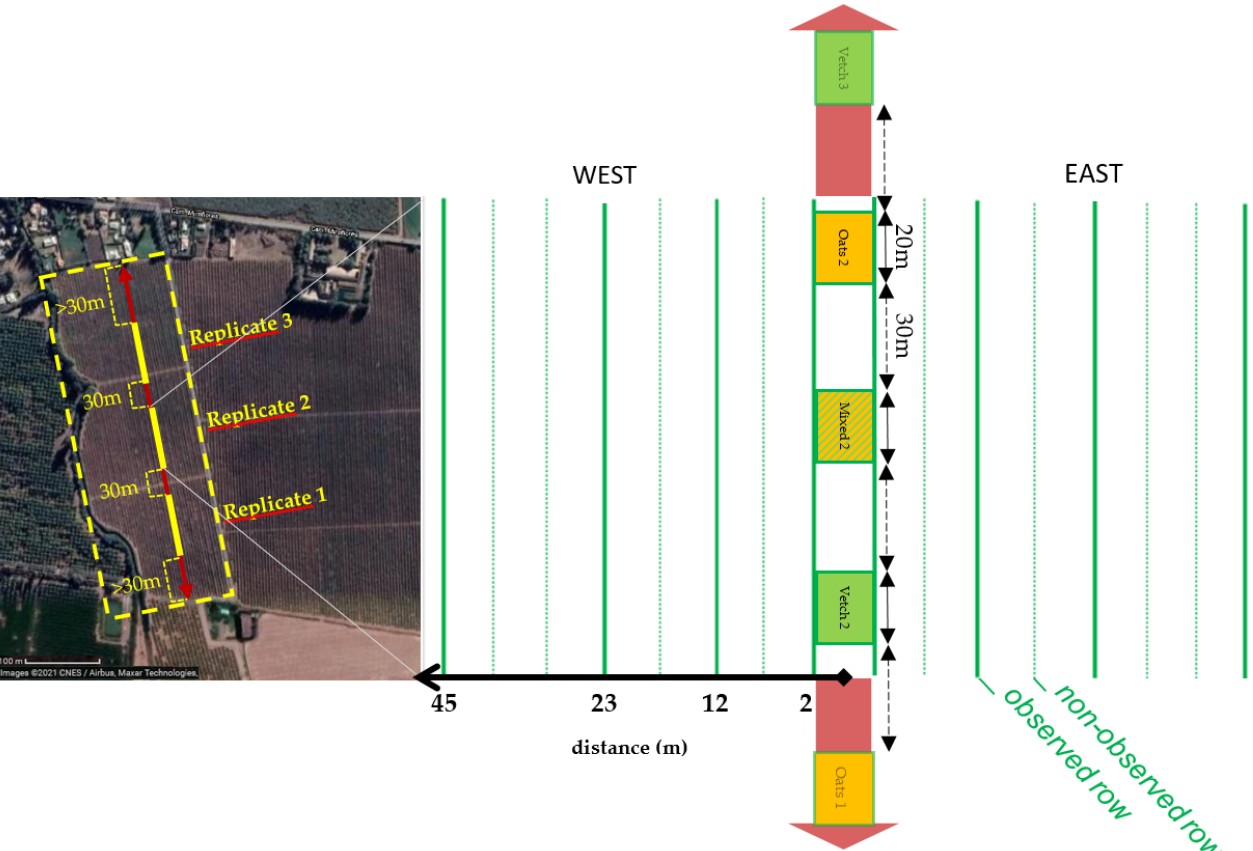

**Figure 1.** Experimental area of the plum orchard indicating the position of the central inter-row (*red arrow*) and the three replicates. The layout of the intercropping treatments are indicated for replicate 2 (same layout as for the other replicates).

*2.3. Trophic Resources Provided by the Central Inter-Row*

Once in early spring (21 August) and then every other week from 26 September to 13 November, the extrafloral nectar release was checked in vetch plants of the vetch and mixed patches. All nectaries of 20 randomly selected vetch plants were observed in every patch. If at least one nectary released nectar, the plant was scored as providing nectar. At the same time, aphids (wingless adults, alate and nymphs; aphid mummies were not included) in the vetch, oats, and mixed patches were sampled by hitting a group of 20 randomly selected plants with a wooden stick, collecting all dropped aphids on a whiteboard, and finally placing them with a paintbrush into Falcon© tubes. This collection method was applied five times in every patch. For the mixed patches, two samplings were performed: one on 20 oat plants, the other on 20 vetch plants. Aphids were brought back to the laboratory on the day of collection and were counted and identified. Aphid species described in the literature as potential hosts for parasitoid species recorded on plum aphids were counted as alternative hosts, and all aphids observed regardless of the species were considered as honeydew providers (see Table 1 for species names and category). All aphids were reared either on oats or on vetch for two weeks and checked daily for mummy formation. The formed mummies were then isolated into plastic tubes until parasitoid emergence. Finally, emerged parasitoids were identified.

**Table 1.** Mean number of aphids per species found in the three types of patches of the inter-row, at each observation date. Rp: *Rhopalosiphum padi*; Sa: *Sitobion avenae*; Sm: *Sipha maydis*; Md: *Metopolophium dirhodum*; Sg: *Schizaphis graminum*; Ap: *Acyrthosiphon pisum*; Ac: *Aphis craccivora*; Mp: *Megourae viciae*; Bh: *Brachycaudus helichrysi*; Af: *Aphis fabae*.

| Observation Date | Intercrop Modality | Rp * | Sa | Sm | Md | Sg * | Ap | Ac * | Mp * | Bh | Af * |
|---|---|---|---|---|---|---|---|---|---|---|---|
| 11 July | Oats | 13.6 | 0.8 | 0.07 | 0.27 | 0 | 0 | 0 | 0 | 0 | 0 |
|  | Mixed | 10.8 | 1.13 | 0 | 0.27 | 0 | 0 | 0 | 0 | 0 | 0 |
|  | Vetch | 0.13 | 0.33 | 0 | 0 | 0.2 | 0 | 0 | 0 | 0 | 0 |
| 21 August | Oats | 0 | 0 | 0 | 0 | 0 | 0 | 0 | 0 | 0 | 0 |
|  | Mixed | 0.53 | 0.13 | 0 | 0.07 | 0.07 | 0 | 0 | 0 | 0 | 0 |
|  | Vetch | 0 | 0 | 0 | 0 | 0 | 0.07 | 0.93 | 0 | 0 | 0.7 |
| 26 September | Oats | 7.67 | 2.47 | 2.93 | 4.47 | 0 | 0 | 0 | 0 | 0 | 0 |
|  | Mixed | 1.13 | 0.47 | 0 | 1.1 | 0 | 0.5 | 0.23 | 0.13 | 0.23 | 0.03 |
|  | Vetch | 0 | 0 | 0 | 0 | 0 | 2.1 | 1.53 | 0.47 | 0 | 0.73 |
| 10 October | Oats | 1.47 | 0.07 | 7.87 | 1.33 | 0.4 | 0 | 0 | 0 | 0 | 0 |
|  | Mixed | 1.83 | 0.57 | 0.97 | 2.1 | 1.3 | 0.6 | 0 | 0 | 0 | 0.07 |
|  | Vetch | 0.13 | 0 | 0 | 0 | 0 | 4.33 | 1.33 | 0.27 | 0.07 | 0.13 |
| 29 October | Oats | 1.13 | 6.27 | 0.73 | 1.07 | 0.6 | 0 | 0 | 0 | 0 | 0 |
|  | Mixed | 0.77 | 1.33 | 1.9 | 0.33 | 0.73 | 1.57 | 0.43 | 0 | 0 | 0 |
|  | Vetch | 0 | 0 | 0 | 0 | 0 | 17.87 | 1.13 | 0 | 0 | 0 |
| 13 November | Oats | 0.2 | 16.27 | 1.67 | 0.07 | 0 | 0 | 0 | 0 | 0 | 0 |
|  | Mixed | 0.13 | 4.77 | 0.37 | 0.07 | 0 | 0.8 | 0 | 0 | 0 | 0 |
|  | Vetch | 0 | 0 | 0 | 0 | 0 | 13.27 | 0.13 | 0 | 0 | 0 |

* Species marked with an asterisk were considered as alternative host species.

### 2.4. Estimation of Aphid Incidence

Aphid incidence on plum trees was estimated on both sides of the central inter-row, on sections of seven trees from the first, third, fifth and eighth plum rows parallel to the central intercropped patches. The corresponding distances were 2 m, 12 m, 23 m and 45 m from the centre of the inter-row, the farthest distance being considered as a negative control. For each distance, on each side of each patch, three of the seven plum trees of the section were randomly selected and the presence or absence of an aphid colony in ten randomly selected shoots from each tree was noted. An aphid colony was considered to be present when at least two aphids were found on the same shoot, regardless of the developmental stage. Aphids were always found on the terminal parts of the shoots, most often at the base of young apical leaves. Aphid incidence on plum trees was assessed four times during the season: 10 and 29 October, 13 November and 4 December.

### 2.5. Parasitism Rate

The parasitism activity in the aphid colonies on plum trees was estimated four times during the season: on 10 and 29 October and on 13 and 27 November. For each plum section at the four distances mentioned above, and on each side of the central inter-row, five aphid colonies were searched for at the edge of the plum tree's shoots. All seven plum trees of the section were observed and, when possible, the five colonies were located on five different trees. For each colony, the species was identified using a hand-held magnifying glass (magnifier capacity ×10) and the number of individuals was counted. Predators and adult parasitoids found on the shoot near the aphid colony were removed, as well as any already formed aphid mummies. Then, an exclusion cage, consisting of a 40 cm long, 20 cm diameter cylinder with mesh fabric, closed at one end with an elastic band to prevent insects from escaping, was placed to cover the entire colony. The other end was closed around the shoot with an iron shaft. The exclusion cages were left for eight days to allow all parasitized aphids to transform into mummies, after which all shoots with exclusion cages containing mummies were cut off and brought back to the laboratory. All mummies

found were counted and isolated in plastic tubes until they emerged, and the emerged parasitoid species were identified following the keys of Starý et al. (1995) and Tomanovic et al. (2014) [52,58]. Parasitism rates were calculated as the ratio of the number of mummies found after 8 days in the colony to the initial number of aphids in the colony when placing the exclusion cage. Only dates for which at least one exclusion cage in every plum section could be placed and showed parasitism activity were considered for analyses, therefore parasitism rates on October 10 were not presented.

*2.6. Statistical Analyses*

All statistical analyses were conducted with the RStudio software (R Core Team, v.3.5.3).

2.6.1. Resources Provided in the Inter-Row

For every observation date, the influence of the patch type (=intercrop) on the abundances of alternative hosts and honeydew providers within the central inter-row was assessed using a generalized linear mixed model (GLMM) assuming a quasipoisson distribution to cope with the overdispersion of the data, with function glmmPQL of package 'MASS'. Models included the patch type as a fixed effect and the replicate as a random effect. A type 3 test was performed using the function Anova of package 'car' to assess the significance of the effects. The same modelling procedure was applied to parasitized aphids collected in the inter-row, but with all observation dates included and considering the observation date as a random effect.

2.6.2. Aphid and Parasitism Incidences and Parasitism Rates

For every observation date, the influences of the distance and the patch type and of their interaction on aphid incidence (proportion of shoots infested by aphids) or parasitism incidence (proportion of cages with mummies) or parasitism rate (ratio previously explained) were assessed using GLMM. Due to a generally strong overdispersion of the data, all models were fitted using function glmmPQL with quasibinomial distribution. Distance and intercrop along with their interaction were considered as fixed effects, and the orientation including the replicate was considered as random variables to account for site effects. Type 3 Anova tests were then performed to assess the significance of the effects. After the GLMM for which type 3 Anova tests indicated a significance of one or both of the fixed effects at the 0.05 level, pairwise comparisons tests were realized with function emmeans of "emmeans" package. When the effect of the interaction was significant (0.05 level), additional GLMMs were realized with subsets of the general dataset including only two types of patches at a time, without changing any other parameter in the models, in order to do pairwise comparisons.

## 3. Results

### 3.1. Availability of Resources in the Inter-Row

From 26 September to 29 October, extrafloral nectar release was continuous in vetch and mixed intercrops: at least one nectary on each plant observed provided extrafloral nectar. Prior to this date, plants had not grown sufficiently and the extrafloral nectaries had not yet formed. During the 13 November observation, plants had completely dried out and the release of extrafloral nectar had stopped.

During the season, up to ten aphid species visited oats and vetch plants in the intercrops, and their abundance varied (Table 1). The collected aphids that turned into mummies belonged mainly to the species *Rhopalosiphum padi* in oats, mostly present at the beginning of the season, and later on to the species *Acyrthosiphon pisum*. On every date, alternative hosts were present, on both oats and vetch, in the three patch types (Figure 2a). Only once on 26 September did the average exceed 5 aphids per 20 plants in the oats, which was significantly higher than in the vetch or the mixed intercrops (Table 2). There was no other significant difference in alternative host abundance between types of patches in the season. Honeydew providers (i.e., total numbers of aphids recorded) were more numerous than

alternative hosts, as logically expected, and they showed more differences in several dates (Figure 2b, Table 2). On 26 September, aphids were over three times more abundant in oats compared to the other two patch types, whereas on 29 October vetch had the highest abundance of aphids. Finally, on 13 November, vetch and oats presented the highest abundance in honeydew providers. The abundance of aphids in the mixed intercrop remained amongst the lowest in all observed dates, with less than 10 aphids per 20 plants on average (all details on pairwise comparison tests are available on Supplementary Materials: Table S1 for alternative hosts and Table S2 for honeydew providers).

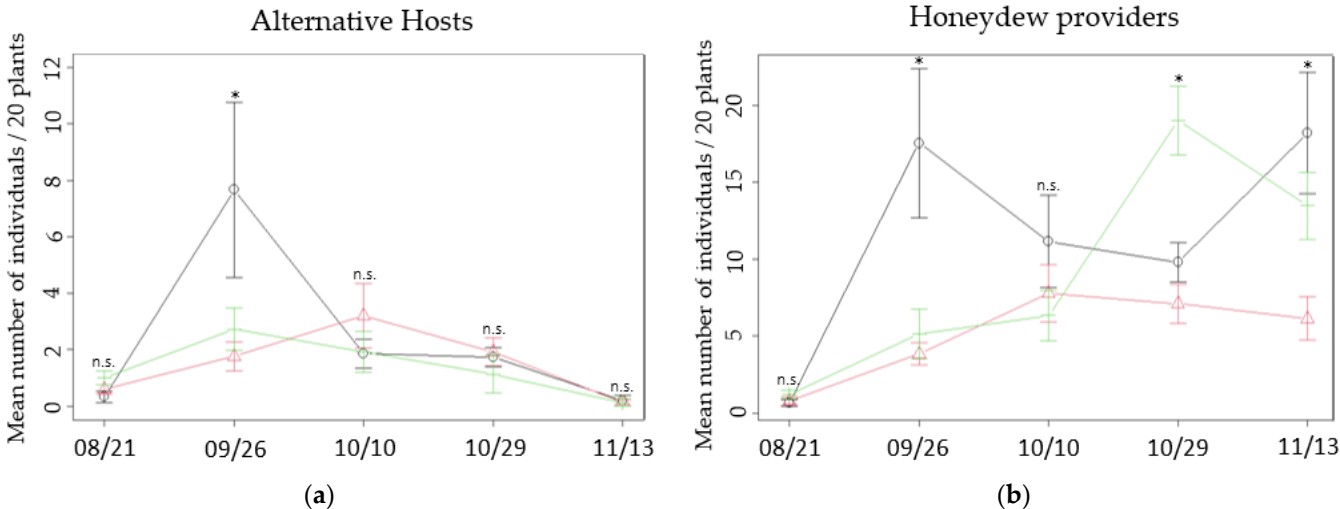

**Figure 2.** Mean number of aphids found per resource type: (**a**) alternative hosts and (**b**) honeydew providers, in the three patch types and at each observation date. Black continuous line: oats; green dotted line: vetch; red dashed line: mixed. Error bars represent standard errors. Significant differences are marked with an asterisk (n.s.: non significant).

Overall, both aphid resource types, alternative hosts and honeydew providers, showed high variabilities in the abundances, on each observation date. Still, the main difference between the intercrops in terms of provided resources was the presence of nectar in the intercrops including vetch plants (vetch and mixed). The main difference between the vetch and mixed patches was therefore the higher available quantity of nectar per patch in the vetch intercrop due to the doubly higher density of vetch plants compared with the mixed intercrop.

Numbers of mummies formed after the aphids were collected in the inter-row presented quite different results from one date to another in the three different types of patches (Figure 3): there was no significant difference in the number of parasitized aphids between intercrops during the whole experimental season (Chisq = 0.0995; Df = 2; *p*-value = 0.951). Parasitism was numerically higher in oats on 26 September and 13 November, whereas in October a higher number of mummies formed was recorded in the patches that included vetch plants. The mixed intercrop was not associated with a higher number of mummies at any date but had the lowest variability in parasitized aphids during the season. Emergence rates were quite low: parasitoids emerged from 24%, 35% and 52% of the total mummies isolated respectively in oats, mixed and vetch intercrops. Emerged parasitoid species included targeted species such as *Aphidius platensis* Brethes and *Lysiphlebus testaceipes* Cresson [Hymenoptera: Braconidae], which attack some aphid species from the inter-row as well as plum tree aphids (Table 3). *A. platensis* emergence occurred earlier in the season than *L. testaceipes* and exclusively from oats and mixed patches, whereas *L. testaceipes* was found on the three types of patches.

**Table 2.** Summary of statistical modelling and of comparisons of intercrops for alternative hosts and honeydew providers in the inter-row. Significant differences (*p*-value < 0.05) are bolded.

| Date | Explained Variable | Fixed Effect (Random Effect) | ANOVA Test Following GLMMs (Df = 2) | | Pairwise Comparisons [1] (Emmeans) *p*-Values (Df = 55) | | |
|---|---|---|---|---|---|---|---|
| | | | Chisq | *p*-Value | Oats/Mixed | Oats/Vetch | Mixed/Vetch |
| 21 August | Alternative hosts | Intercrop (replicate) | 5.1441 | 0.076 | / | / | / |
| | Honeydew providers | Intercrop (replicate) | 2.4817 | 0.289 | / | / | / |
| 26 September | Alternative hosts | Intercrop (replicate) | 12.132 | **0.002** | **0.002** | **0.046** | 0.453 |
| | Honeydew providers | Intercrop (replicate) | 31.136 | **<0.001** | **<0.001** | **0.001** | 0.473 |
| 10 October | Alternative hosts | Intercrop (replicate) | 1.301 | 0.522 | / | / | / |
| | Honeydew providers | Intercrop (replicate) | 2.0423 | 0.360 | / | / | / |
| 29 October | Alternative hosts | Intercrop (replicate) | 1.0802 | 0.583 | / | / | / |
| | Honeydew providers | Intercrop (replicate) | 26.000 | **<0.001** | 0.187 | **0.005** | **<0.001** |
| 13 November | Alternative hosts | Intercrop (replicate) | 0.178 | 0.915 | / | / | / |
| | Honeydew providers | Intercrop (replicate) | 14.847 | **<0.001** | **<0.001** | 0.295 | **0.015** |

[1] Pairwise comparison tests were not performed when Anova test *p*-values were not significant.

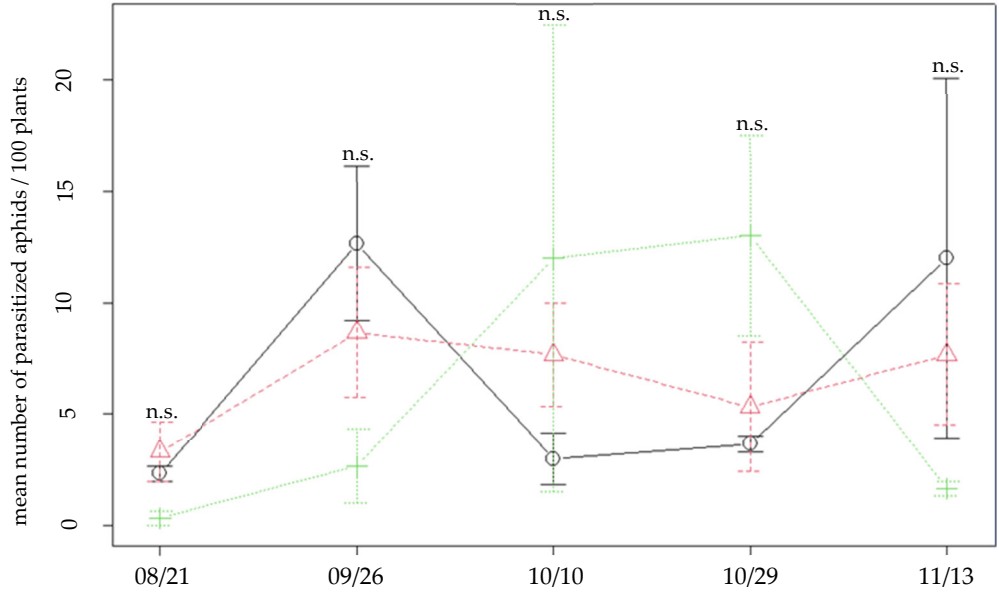

**Figure 3.** Mean number of parasitized aphids after every sampling in the inter-row, for all three patch types. Black continuous line: oats; green dotted line: vetch; red dashed line: mixed. Error bars represent standard errors. Significant differences are marked with an asterisk (n.s.: non significant).

**Table 3.** Total number of parasitoids (and hyperparasitoids) per category from collected aphids of all three patch types. Ae: *Aphidius ervi*; Ap: *Aphidius platensis*; Au: *Aphidius uzbekistanicus*; Lt: *Lysiphlebus testaceipes*; Pg: *Praon gallicum*; Pv: *Praon volucre*; P sp.: *Praon* sp.; H: Hyperparasitoids; NI: Not identifiable; NE: Non-emerged.

| Date | Intercrop Modality | Ae | Ap | Au | Lt | Pg | Pv | P sp. | H | NI | NE |
|------|--------------------|----|----|----|----|----|----|-------|---|----|----|
| | Oats | 0 | 3 | 0 | 0 | 0 | 0 | 0 | 1 | 2 | 1 |
| 21 August | Mixed | 1 | 3 | 0 | 0 | 0 | 0 | 0 | 0 | 2 | 4 |
| | Vetch | 1 | 0 | 0 | 0 | 0 | 0 | 0 | 0 | 0 | 0 |
| | Oats | 0 | 1 | 0 | 2 | 0 | 0 | 0 | 0 | 0 | 35 |
| 26 September | Mixed | 0 | 1 | 0 | 0 | 0 | 0 | 0 | 0 | 0 | 25 |
| | Vetch | 0 | 0 | 0 | 0 | 0 | 0 | 0 | 0 | 0 | 8 |
| | Oats | 0 | 1 | 0 | 0 | 0 | 0 | 0 | 1 | 5 | 2 |
| 10 October | Mixed | 7 | 1 | 0 | 0 | 4 | 1 | 0 | 0 | 4 | 6 |
| | Vetch | 2 | 0 | 0 | 8 | 0 | 0 | 0 | 5 | 1 | 20 |
| | Oats | 0 | 0 | 1 | 0 | 0 | 1 | 0 | 0 | 2 | 7 |
| 29 October | Mixed | 1 | 0 | 0 | 7 | 0 | 0 | 0 | 0 | 1 | 7 |
| | Vetch | 23 | 0 | 0 | 7 | 0 | 0 | 0 | 0 | 1 | 8 |
| | Oats | 1 | 0 | 10 | 0 | 0 | 0 | 4 | 0 | 12 | 9 |
| 13 November | Mixed | 0 | 0 | 8 | 0 | 0 | 0 | 0 | 0 | 6 | 9 |
| | Vetch | 3 | 0 | 0 | 0 | 0 | 0 | 0 | 0 | 1 | 1 |

*3.2. Aphid Incidence*

No clear effect of the distance from the inter-row was detected on aphid incidence in plum trees, at any observation date (Figure 4, see Supplementary Materials Table S3). Still, on November 13, the effects of intercrop and its interaction with the distance were significant (effect of the intercrop: Chisq = 7.5128; Df = 2; *p*-value = 0.023; effect of the interaction: Chisq = 10.577; Df = 2; *p*-value = 0.005). On this date, aphid incidence in oats-associated plum trees was lower than for the other two patch types, more particularly when closest to the patch (see Supplementary Materials Table S4 for the model summary). At 2 m distance, the infestation by aphids was barely above 1% for oats-associated plum trees, whereas it reached around 6% and 10% for vetch and mixed-associated plum trees respectively. However, proportions of infested shoots were quite variable. On average, the proportions of infested shoots remained low throughout the season and at all distances, the maximum (10% on average) being reached by the mixed-associated plum trees, at the closest distance from the strip, on November 13 (Figure 4c).

*3.3. Incidence of Parasitism and Parasitism Rate*

Plum aphid mummies found in the exclusion cages were identified as either *B. helichrysi*, *Brachycaudus persicae* (Passerini), *Aphis spiraecola* Patch or *M. persicae* [Hemiptera: Aphididae], four species known to attack plum trees and other fruit trees. The incidence of parasitism amongst exclusion cages sometimes reached high levels, with a mean incidence of 70% of the cages with at least one aphid mummy in the mixed intercrop on November 13, closely followed by the vetch intercrop (Figure 5). There was however no significant effect of the distance from the central inter-row, or of the type of patch, or their interaction, on the incidence of parasitism amongst exclusion cages on any observation date (for statistic results, see Supplementary Materials Table S3). Yet, irrespective of the location in the orchard, parasitism was omnipresent, especially in late October and November.

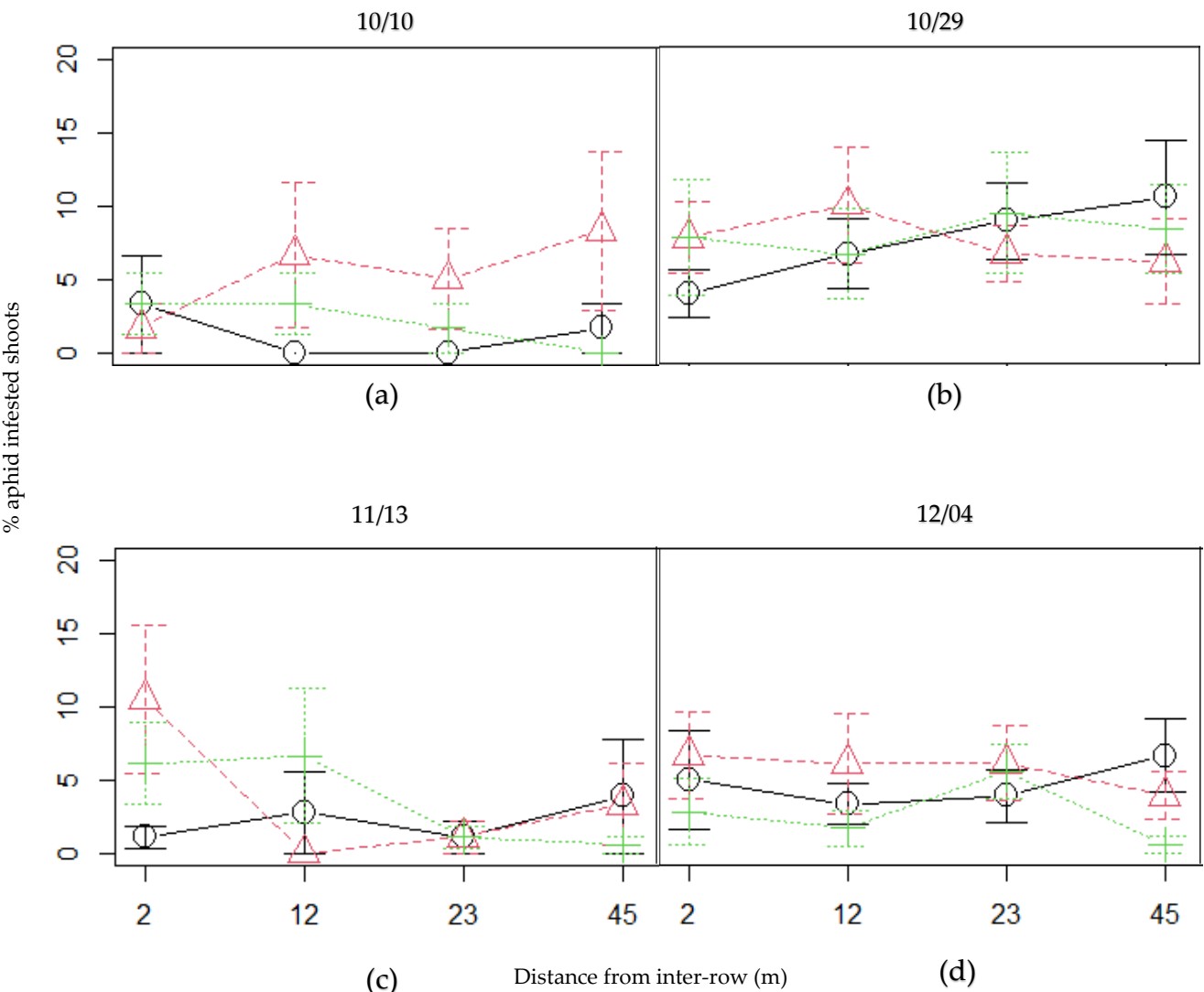

**Figure 4.** Proportion of plum tree shoots infested by aphids at four distances from every patch type, for every observation date: (**a**) October 10, (**b**) October 29, (**c**) November 13 and (**d**) December 4. Black continuous line: oats; green dotted line: vetch; red dashed line: mixed. Error bars represent standard errors.

Resulting parasitism rates in plum tree aphids were quite high on 29 October and 13 November, reaching almost 60% in the plum trees closest to the mixed patches (Figure 6). Although there was no difference between patch types in terms of parasitism incidence, parasitism rates on 29 October and 27 November significantly differed according to the patch type (effect of the intercrop on 29 October: Chisq = 6.6492; Df = 2; *p*-value = 0.036; on 27 November: Chisq = 6.7004; Df = 2; *p*-value = 0.035). On 29 October, the parasitism rate in oats-associated plum aphids was about half that of the other two patch types, and on 27 November the vetch-associated plum aphids presented a higher parasitism rate than the mixed-associated plum aphids (see Supplementary Materials Table S5 for pairwise comparison tests details). On 13 November, parasitism rates were not significantly different between patch types or distances (see Table S3 for a summary of statistic results).

Similarly to mummies from collected aphids in the inter-row, emergence rates in mummies from collected plum aphids were too low for quantitative comparisons. However, the two main parasitoid species that emerged were *A. platensis* and *L. testaceipes*, which were also found emerging from mummies in the inter-row (Figure 7).

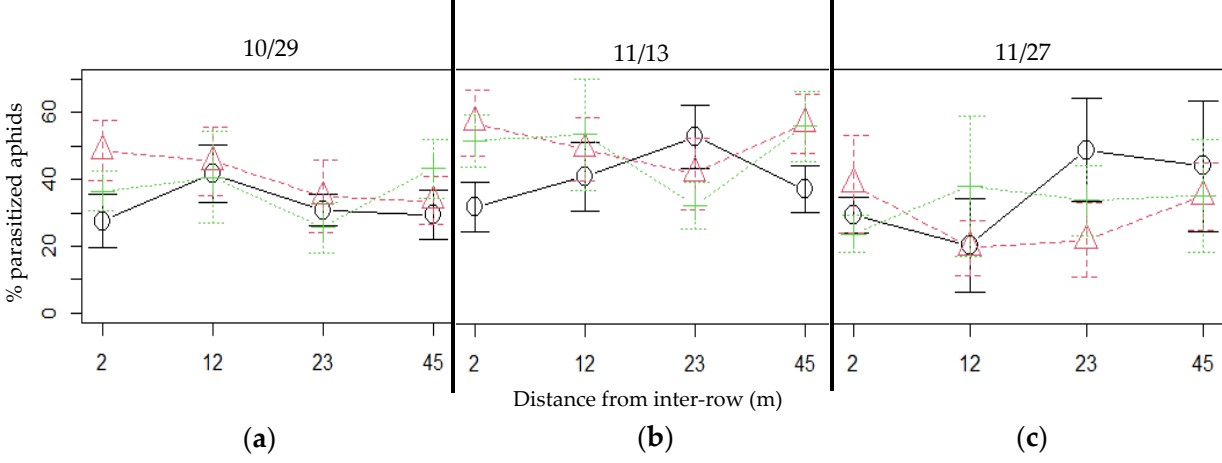

**Figure 5.** Incidence of aphid parasitism during the observational season in the plum trees, at four distances from every patch type, on all observation dates: (**a**) October 10, (**b**) October 29, (**c**) November 13 and (**d**) December 4. Black continuous line: oats; green dotted line: vetch; red dashed line: mixed. Error bars represent the standard error.

**Figure 6.** Parasitism rates in the plum trees at four distances from each patch type, on three observation dates: (**a**) October 29, (**b**) November 13 and (**c**) November 27. Black continuous line: oats; green dotted line: vetch; red dashed line: mixed. Error bars represent the standard error.

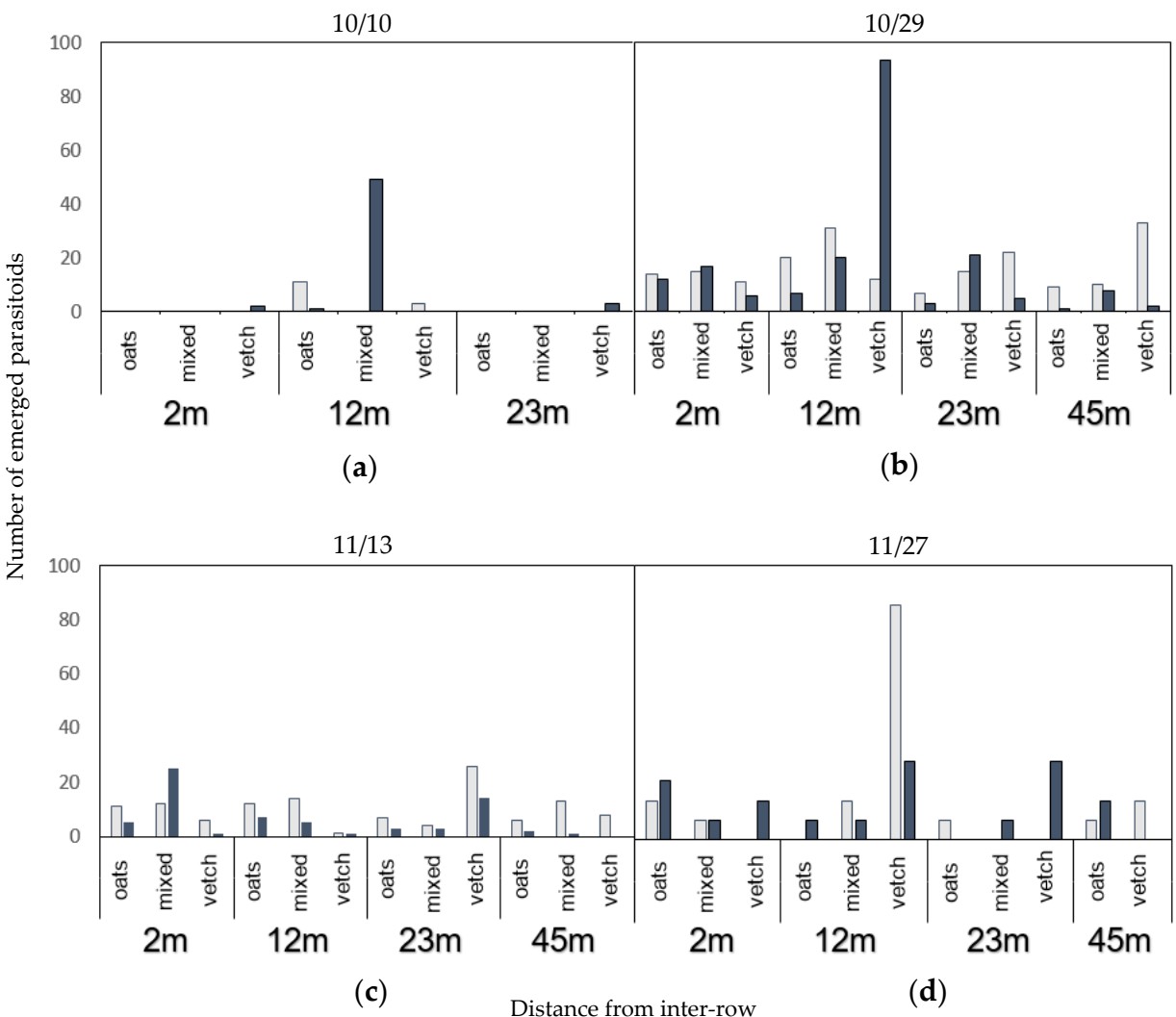

**Figure 7.** Numbers of emerged *A. platensis* (bright) and *L. testaceipes* (dark), from each exclusion cage collection in plum trees for all types of patches and distances, per experimental date: (**a**) 10 October, (**b**) 29 October, (**c**) 13 November and (**d**) 27 November. No parasitoid emerged from mummies collected on 10 October at 45 m.

## 4. Discussion

### 4.1. Abundance, Quality and Temporality of Carbohydrates Provided in the Inter-Row

Although intercrops were well established within the plum orchard, vetch density in the mixed and vetch intercrops was lower than expected, probably due to a low germination rate (*personal observation*). Still, the delivery of extrafloral nectar by vetch plants started at the early stages of plant development with the formation of the stipules in late August (early spring in Chile) and ended when plants started to dry by the end of November. Extrafloral nectar release is subject to abiotic parameters [21]. Mild and warm temperatures and sunny weather occurred during most of the 2019 season and allowed the continuous release of extrafloral nectar by the great majority of vetch plants. It is therefore improbable that extrafloral nectar availability was limiting in the vetch and mixed patches for parasitoids. Honeydew availability varied during the season with total aphid abundances and was higher in oats at the beginning and higher in vetch at the end. Mixing cereals and leguminous plants has been shown to reduce the incidence of some aphids by bottom-up control [59–62]. It could explain why there were fewer aphids in the mixed intercrop, even if

our initial hypothesis was rather to attract a higher number of parasitoids with the presence of hosts, honeydew and extrafloral nectar combined and produce a top-down regulation.

Although honeydew and extrafloral nectar were available, we could not disentangle in this experiment whether parasitoids fed in the intercrops, and if they did, did they prefer one or other sugar resource. The fact that both resources were available continuously in the vetch and mixed patches, when only honeydew was available in the oats patches, was the expected outcome of carbohydrates resources from the inter-row. However, a possible drawback of providing sugar sources, recently suggested by Heimpel [63], is that fed parasitoids are encouraged to disperse further away from the carbohydrate sources as a strategy to avoid density-dependent hyperparasitoid attraction, self-superparasitism, inbreeding among offspring and catastrophic mortality events. If this hypothesis is confirmed, there would have been an effect of intercropping, but the spatial scale considered in this study would probably have been too small to detect it.

### 4.2. Abundance, Quality and Temporality of Alternative Hosts and Parasitism in the Inter-Row

The other resource provided by the three intercrops included alternative aphid hosts. These were more abundant in the oats intercrop in late September which could be due in part to the oats germinating and growing earlier than vetches [64]. A higher proportion of parasitized aphids in the oats at the same time is consistent with the highest proportion of alternative hosts, suggesting that cereal alternative hosts did attract parasitoids early in the season, which is favourable for successful biological control of the aphid pests later in the season [9,65,66]. Among the species found parasitizing the inter-row aphids, *Aphidius platensis* and *A. uzbekistanicus* were represented, however, their host ranges do not include any of the aphid species found on vetch plants, which explains why they were only found in oats and mixed intercrops [52,67]. Unlike these two species, *Aphidius ervi* is known to prefer legume-feeding aphids such as *Acyrthosiphon pisum* [68]. Vetch and oat aphids possibly attracted different parasitoid species at different times and densities, which could explain the differences in parasitized aphids. This result partially confirms our assumption that oats and vetch effectively provided alternative hosts at two different periods of the season. Nevertheless, the number of parasitized aphids was unexpectedly never higher in the mixed intercrop, which suggested that, unlike what has been observed by Jamont et al. [26], parasitoids were not mainly attracted by a habitat gathering alternative hosts and carbohydrates sources but that other dynamics drove them in one way or the other to intercrops [61].

Alternative host abundance in the mixed intercrop was not higher than in the other two, despite harbouring a higher diversity of aphid species, which is possibly linked with the effect of intercropping on aphid incidence, as previously mentioned.

### 4.3. Dispersal of Parasitoids between the Intercrops and the Plum Trees

The two main parasitoid species observed parasitizing plum aphids, *Aphidius platensis* and *Lysiphlebus testaceipes*, were also identified as species parasitizing alternative hosts in the intercrops. *Aphidius platensis* was more associated with oats and mixed intercrops, whereas most *L. testaceipes* individuals were found in vetch and mixed intercrops. No clear effect of the vicinity of the intercropping on parasitism incidence in plum aphids was detected and, unfortunately, parasitoid emergence was too low to be able to determine whether one species or another was more abundant in plum trees associated with one of the three intercropping modalities. Nevertheless, the fact that we found shared parasitoid species between the two vegetation levels may indicate that transfer between them is possible, which has recently been observed by Alvarez-Baca et al. (2020) on *A. platensis* between cereal aphids and *Myzus persicae* aphids in peach orchards [54]. Our methodology was possibly insufficient to validate the field dispersal of these organisms, as parasitoids movements were not followed in the field, directly or indirectly. And yet, even if some parasitoid species have been observed in the intercrops and in the plum

trees, dispersal between these two compartments could have been restricted due to host or habitat specialization [69,70], or host fidelity with or without genetic local adaptation [68].

Results from our study indicate that parasitoids species were found in common between aphids from the intercrops and plum aphids. However, the fact that we detected no clear effect of the vicinity of the inter-row strips on aphid or parasitism incidences in plum trees suggests little or no movement between plum trees and intercrops.

### 4.4. Potential Negative Effect of the Inter-Row on Parasitism in Plum Trees

There was no significant difference in parasitism incidence among the treatments, either between intercrops and plum trees or among intercrops, even at 2 m from the inter-row. One significant result alone cannot be considered as optimized for a successful parasitism activity, given the number of models that were used to test parameters, and given the complexity of host selection in parasitoids (already mentioned). A possible setback effect of intercropping was that parasitoids may have been retained within the intercrops and prevented from leaving it to forage for plum aphids, since some of their preferred hosts were available closer to the extrafloral nectar source during the season [17,26,32]. Alternatively, there might have been a barrier effect between sections so that parasitoids were easily able to move between intercrops rather than switching to the plum tree compartment. Inter-row sections were 20 m wide, and the area of intercropping was rather small compared to the orchard area, possibly making it difficult to detect any clear effect it may have had on plum trees. On the other hand, there is no direct data on the movements of the main parasitoid species common to both intercrops and plum trees. The dispersion of these parasitoid species may be underestimated [71].

### 4.5. Plum Aphids Incidence and Their Regulation

The incidence of plum aphids did not clearly differ with either the intercrop or the distance from the inter-row, although there was an effect at one sampling period when inter-row sections had started to dry and fewer alternative aphid hosts were observed in the inter-row. As previously mentioned, local adaptation, habitat specialization or host fidelity in parasitoid species may have prevented migration between both compartments. It is also possible that, as suggested by Luquet et al. [72], the honeydew released by plum aphids was sufficient for parasitoids, and more preferentially exploited than the extrafloral nectar from the inter-row, since honeydew is available directly where hosts are.

In this experiment, predation was not taken into account. The exclusion cages estimated optimized parasitism rates without considering intraguild predation. By their intraguild action on parasitized aphids, predators could have affected the outcomes on parasitism. Alternatively, some predators have been observed to preferentially consume non-parasitized aphids, in which case predators could have decreased the number of non-parasitized aphids, as competitors of parasitoids [73]. The extent of predation may have been valuable information to understand the low aphid incidences [74], since we observed many aphid predators such as Coccinellidae in the orchard during the season [*personal observation*]. Generalist predators do not exhibit host fidelity, they move over larger distances than parasitoids and could have dispersed in the whole orchard, preventing us from properly interpreting the effect of the inter-row on aphid incidence [75]. Generalist predator species present during this study exhibit greater diet breath and other inter-annual dynamics, which have been shown to be highly dependent on the landscape structure [76]. Therefore, they could be affected by other alternative hosts and nectar provisions in the agrosystem under study [77–79].

### 4.6. Limitations of the Study

This experiment was realized over one single growing season, which can bring doubts regarding the generalization and the legitimacy of the results. For instance, it could be argued that aphid or parasitoid incidences this particular year were not representative. However, previous similar experiments realized in the same plum orchard have also

estimated the aphid incidence the year before: aphid incidence nearby an oats-filled inter-row was similar to the average aphid incidence in plum shoots found in 2019 at the beginning of November (Alvarez-Baca et al., in prep.).

The number of replicates was only three but there was a trade-off between the size of the intercrop treatments and the distance between replicates, and given previous field experiments searching for parasitoid movements from provided resources [17,80], the experimental design was chosen to maximize intercrop strip length as well as the distance between them, to have enough intercrop area to observe a possible effect but also to avoid possible overlaps and interference between treatments. Three replicates may have generated low statistical power in order to detect some weak effects of the intercropping modalities. In any case, the context-dependency of such pest management practices has been reported before [81]. Additional experimentations testing the complementarity of conservation biological control practices such as the use of banker or nectariferous plants would be needed to interpret more accurately the possible beneficial role of intercropping in plum orchard inter-rows for biological control of aphids.

## 5. Conclusions

This study emphasizes the context-dependent efficiency in the use of intercrops to provide resources needed by parasitoids for biological control, and highlights that providing two types of trophic resources or combining two types of practices, does not systematically bring higher efficiency in parasitoid recruitment or activity [82]. Here, mixing cereal and leguminous plants did not seem to bring an additional or synergetic effect on aphid control or parasitoids when compared with the same crops provided separately. Nevertheless, this study does validate some of the assumptions made by providing trophic resources with intercropping, but without leading to increased biological control of pests. The companion plant types implanted in the central inter-row inside the plum orchard experimental area provided the expected resources, that is alternative hosts and extrafloral nectar, but had no clear outcome on the regulation of plum aphids by parasitoids. Even so and without giving quantitative information, our study helps to illustrate the communities of parasitoids that can be found and potentially shared between two completely different crop types. It highlights the necessity to investigate wider trophic webs than plant–aphid–parasitoid complexes, in order to efficiently develop optimized pest management programs in orchards thanks to intercropping or other environmental-friendly practices.

**Supplementary Materials:** The following are available online at https://www.mdpi.com/article/10.3390/agronomy12010077/s1, Table S1: pairwise comparison results for GLMM testing the effect of intercropping modality on the abundance of alternative hosts in the inter-row on 26 September; Table S2: pairwise comparison results for GLMM testing the effect of intercropping modality on the abundance of honeydew providers in the inter-row on 26 September, 29 October and 13 November; Table S3: results summary of generalized linear mixed modelling of the effects of distance, intercrop modalities and their interaction on aphid and parasitism incidences and parasitism rate, in plum trees; Table S4: pairwise comparison results for GLMMs testing the effect of intercropping modality and distance interaction on aphid incidence in plum trees on 13 November; Table S5: pairwise comparison results for GLMM testing the effect of intercropping modality on parasitism rate in plum aphids on 29 October and 27 November.

**Author Contributions:** Conceptualization, L.L., B.L., B.J. and M.P.; methodology, L.L., E.C., C.M.V., N.C., B.L., B.J. and M.P.; validation, M.P., B.J., B.L. and L.L.; formal analysis, L.L., B.L. and M.P.; investigation, L.L., E.C., C.M.V., N.C., B.J. and B.L.; resources, L.L., E.C., C.M.V. and N.C.; data curation, L.L.; writing—original draft preparation, L.L.; writing—review and editing, L.L., B.J., B.L. and M.P.; visualization, L.L. and E.C.; supervision, B.J., B.L. and M.P.; project administration, B.J. and B.L.; funding acquisition, B.L., B.J. and M.P. All authors have read and agreed to the published version of the manuscript.

**Funding:** This research was funded by Fondo Nacional de Desarrollo Científico y Tecnológico (FONDECYT) regular grant n°1180601 to B.L. The authors would also like to thank project ANID/PIA/ACT192027 for funding provided during the writing of this manuscript. L.L. was financially supported by the Contrat Plan État Région (CPER) Pays de la Loire (2015–2020) and her mobility was funded by an Erasmus + International Credit Mobility grant supported by project convention 2018-1-FR01-KA107-046996.

**Institutional Review Board Statement:** Not applicable.

**Informed Consent Statement:** Not applicable.

**Data Availability Statement:** The datasets used and/or analyzed and presented in this study are available from the corresponding author by mail request.

**Acknowledgments:** The authors thank Jeniffer Alvarez-Baca, Armando Alfaro-Tapia, Macarena Guiachetti, Luciano Hidalgo, Leonardo Cifuentes and Washington Saldías for their help in the field. They also express their gratitude and special thanks to Nicolás Gonzales, Teresa Vallejos and their agronomic team for their availability and support in the plum orchard during the whole experimental time.

**Conflicts of Interest:** The authors declare no conflict of interest.

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
