# Peer review of "Providing Alternative Hosts and Nectar to Aphid Parasitoids in a Plum Orchard to Determine Resource Complementarity and Distance Range Effect on Biological Control"

_agronomy, doi:10.3390/agronomy12010077_

Round 1

Reviewer 1 Report

The manuscript presents the results of a study examining the role of enhancing alternative hosts and supplementing carbohydrate sources for parasitoids in an orchard system (plums). The work is more holistic than many other similar studies, in that the authors examine both the effects of alternate hosts and nectar on parasitoid presence in the intercrop, as well as the actual impact of the parasitoids in the target crop. I commend the author for this balanced and synthetic work. The paper is generally well-written and clear. Likewise, much of the design is appropriate and the analyses appear to be good.

I have some serious concerns, however, which unfortunately cannot be easily addressed. The first is that the work is a single-season field study. It is very difficult to extract meaningfully predictive data/results from a single field season due to year-to-year variations in weather (and with that, overall cropping system phenology), production practices, and arthropod abundance. It would have been much better to have a second or even third year.

Second, the spatial separation of the three replicates is not mentioned, but is presumably close, based on the aerial image provided. The authors note that there is a 20-meter gap lengthwise between each intercrop plot within a replicate, but distance between replicates is not mentioned. Twenty meters is fairly small, given that the parasitoids are presumably colonizing the orchard from the outside (all non-intercrop areas were mowed, as I understand it) over a greater distance, and thus must be capable of traveling some distance so could be expected to be able to move readily between plots. Further, close proximity of the plots to one another would allow volatile cues to mix between the treatments, potentially confounding parasitoid selection of plots. What was the separation between replicates?

As a result of these design issues, I am unsure that the paper can contribute to the body of science around habitat manipulation for biological control.

Specific Comments:

Line(s)

Comments

108-110

I think I missed the point the authors were trying to make here. If the parasitoids have no preference for oat or plum aphids, why would they be expected to leave the intercrop unless hosts became unavailable there? A preference for another habitat or host type would be key to the parasitoids dispersing, not indifference as the authors suggest.

114

Change “Aphidiinae” to “aphidiine”

115

Change to “nitrogen-fixing legume, vetch does not compete”

128-129

Change to “whether additive or synergistic”

132

Change to “incidence in plum trees will decline”

Fig 1

Were the respective intercrop treatments in the same orientation in each of the three replicates?

157

What was the distance between each experimental plot?

159

Change to “avoid the presence of nectar sources or aphid hosts”

170-171

Were the numbers of plants sampled in the mixed intercrop adjusted to per 20 plants for the combination (the vetch only and oats-only treatments had counts based on 20 plants, but a total of 40 plants – 20 oats and 20 vetch – were sampled in the mixed plots)? Are there plant stand counts for the respective intercrop treatments so that we can see how well the plants established, and can get a better idea of how abundant alternate host pants and nectar sources were? The authors indicate that there was some difficulty in getting a good stand of vetch. What were the actual plant densities, and what would be the ideal density in the vetch-only and mixed stands?

208

Change to “There were insufficient numbers of aphid colonies”

209

Change to “so parasitism rates were not represented”

249-256

Which aphid species are considered alternate? It would be useful to list those species that were on plums only, and those that were designated as alternate hosts.

268

And everywhere else in the manuscript this species is spelled out: Change “Acyrtosiphon” to “Acyrthosiphon”

280

Change “statistic” to “statistical”

294

Change to ”Parasitism was numerically higher in oats”

297-300

Any idea whether host aphid species influenced these low rates of parasitoid emergence? What about hyperparasitoid emergence? Were any observed?

328

Change to “mummies found in the exclusion cages were identified as either”

345

Change to “about half that of the other two intercrops”

417

Change to “were represented, however their host ranges do not include any of the aphid species”

448-453

I’m not sure how the authors decided that the shared parasitoids seemed to visit plum trees independently of the associated intercrop”. It is certainly possible, but the experiment provides no insights on this point. Arguing for low dispersal seems to be a difficult position to defend, as the parasitoids presumably had to travel some distance to initially colonize the intercrop islands in the orchards, or the plum trees. There is the possibility of differential chemical cues between the plums and the intercrops that differentially attract shared parasitoids. Aphid density and relative abundance overall in the intercrops and plums also may have played a role in parasitoid responses to plums relative to intercrops. Further, if aphids were on trees that were between the colonizing parasitoids and the intercrops, it may have distracted those parasitoids from using intercrops as a resource prior to using the plums.

461

Change to “since some of their preferred hosts were available”

470

Change to “genetics data has been suggested between plots kilometres away”

Conclusions

The authors note in the paragraph preceding the conclusions section that predators may have affected the outcomes. The intraguild actions of the generalist predators with the parasitoids could have influenced outcomes, although there is at least one known example of lady beetles preferring to consume unparasitized aphids over parasitized ones (Snyder and Ives, 2003, Ecology 84(1):91-107), which would actually inflate the perceived value of parasitoids.

Author Response

The authors' reply is available in the pdf file.

Reviewer 2 Report

The subject manuscript is an ambitious study attempting to examine the effects of different intercropping strategies on biological control in plum orchards. The authors are reporting mostly negative results, which in my mind is not a bad thing; negative results can be just as informative as positive results, and they should be published more often. The manuscript is generally well-written and clear, and the overall study design is appropriate to address the hypotheses. However, some details of the study design are problematic. First, the low number of replicates (n=3; single field season) means that we cannot be certain if the negative results are “real” phenomena, or merely reflect a lack of statistical power. Second, the study lacks a negative control. No data were apparently collected from plum trees adjacent to mown sections (i.e., no sown intercrop). Therefore, the authors can only evaluate the effects of their different sown intercrops, and not the effect of having an intercrop vs. no intercrop at all. This seems like a substantial missed opportunity, as comparing trees adjacent to an intercrop with trees adjacent to mown ground would seem to offer the best chance for rejecting the null hypothesis.

I have made some additional comments and suggestions on specific parts of the manuscript below. Overall the manuscript is very well-written, with some minor grammar and usage problems throughout. I’ve attempted to suggest corrections for many of these, but my comments should not be considered a comprehensive review of grammar.

Line 14: change “form part of” to “contribute to”

Line 24: change “to” to “with”

Line 25: delete “to”

Line 25: change colon to semicolon or, preferably, start a new sentence with “The context-dependent…”

Lines 28-29: the terms “aphid parasitoids” and “biological control” are both in the title. Do these also need to be in the keywords?

Line 57: change “carbohydrates” to “carbohydrate”

Line 94: Change “The comparison of the effect” to “Comparisons of the effects”

Line 100: Change “Implanting” to “Planting”

Line 102: Delete “needed”

Line 114: change “Aphidiinae” to “aphidiine”

Line 115: change “fixating leguminous” to “fixing legume”

Figure 1: I suggest rescaling the blocks in the figure so that the 30-meter block is a little longer than the 20-meter block. The figure need not be perfectly to scale, but it looks strange to display a distance of 30-meters as shorter than a distance of 20-meters.

Line 157: Were the positions of each crop modality random within each block? Or was the order always Oats (north), mixed (middle), vetch (south)?

Line 208: change “no sufficient” to “insufficient.” Also, explain how it was determined that too few aphid colonies. were found on this date to include in the analysis. In other words, how do you define “insufficient?”

Lines 285-289: Revise for clarity. Maybe break into two sentences.

Line 296: change “to” to “with”

Table 3: What is meant by “not identifiable”? Everything that emerged should be identifiable to some taxonomic level.

Line 310: Change the title of the table to reflect the reality of the data presented (it is not a just a table of emerged parasitoids; it also contains counted of non-emerged parasitoids).

Line 353: change “Figures” to “Figure”

Line 350: I am confused about what exactly was collected and measured here. From Figure 1, we can see that the “inter-row” is the section between plum tree rows, where the intercrops were planted. How, then can aphids “collected in the inter-row” have developed into “plum tree mummies”?

Figure 7c: the words “mixed” and “vetch” are truncated in the x-axis labels.

Line 392: change “among” to “in”

Line 402: change “carbohydrates” to “carbohydrate”

Line 415: change “oats” to “oat”

Line 415: I’m having trouble squaring the statement that “vetch aphids seemed to harbour more parasitoids than oats aphids” with the data in table 3. The statement appears to be true for only two of the five dates (10/10 and 29/10). The total number of parasitoids (including hypers, un-identifiables and non-emerged) is 101 for oats and 89 for vetch. Granted, these numbers are heavily skewed by the counts of non-emerged parasitoids, and the table does not break down how many parasitoids were from oat vs. vetch in the mixed intercrop. However, it does not appear to be generally true, at least from the data as presented, that there were more parasitoids in vetch vs. oat.

Line 417: Change “host” to “hosts”

Line 419: Change “leguminous aphids” to “legume-feeding aphids”

Line 420: Change “oats” to “oat”

Line 428: change “hosts” to “host”

Line 434: avoid starting a sentence with a genus abbreviation.

Line 446: Delete the comma after “restricted”

Lines 450-453: In order to remove the passive voice and improve clarity, sentence can be re-written as, “But the fact that we detected no clear effect of the vicinity of the inter-row strips on aphid or parasitism incidences in plum trees suggests little or no insect dispersal between trees and intercrops.”

Line 455-466: Suggest rewording as “There was no significant difference in parasitism incidence among the treatments, either between intercrops and trees, or among intercrops, even at 2m from the inter-row”

Lines 456-459: To what “significant positive result” are you referring? None of the figures summarizing parasitism show any significantly significant results.

Lines 462-464: I don’t think “diffusion effect” is a good name for this phenomenon. Diffusion suggests passive movement (e.g., down a concentration gradient) when the parasitoids may actively prefer to move among the similar habitat types within the intercropping zone rather than between the intercropping zone and the trees.

Lines 468-470: insert “parasitoid” in front of “species,” and put a period after “underestimated.” The next sentence citing [71] should be more carefully constructed to better-reflect the results of that study. The authors of [17] found no population structure among parasitoid subpopulations, even for subpopulations more than 400km apart. However, that does not mean that individual parasitoids can disperse hundreds of kilometers. The study examines gene flow (which can occur over long distances via many short dispersal events); it does not examine dispersal ability of individual parasitoids.

Line 482: delete “a”

Line 486: change “to properly interpret” to “from properly interpreting”

Line 502: delete “it”

Author Response

(The authors gave the same response as above.)

Reviewer 3 Report

This is an interesting study evaluating the impact of mixed nutritional resources on efficacy of biological control of aphids.  Overall the study design is appropriate in comparing uniform plantings with a mixed planting. 

The only element of design that raises concerns is that at increasing distances there is overlap in range from the various treatments.  This is a common issue in field plots, but it should be considered when taking measurements at increasing distances. 

Author Response

(The authors gave the same response as above.)

Round 2

Reviewer 2 Report

The authors have suitably addressed my concerns.

Author Response

The authors are pleased that they suitably replied to the Reviewer's comments and suggestions.